Psychometric properties of the Attitudes towards Medical Communication Scale in nursing students

Escribano Silvia 1
Juliá-Sanchis Rocío rjulia@ua.es 1
García-Sanjuán Sofía 1
Congost-Maestre Nereida 2
Cabañero-Martínez María José 1
1 Faculty of Health Science, Department of Nursing, University of Alicante , San Vicente del Raspeig , Alicante , Spain
2 Department of English Studies, Faculty of Arts, University of Alicante , San Vicente del Raspeig , Alicante , Spain
Moran-Garcia Jose Maria
Electronic publication date: 2021 May 25
Publication date: 2021
Volume: 9
Electronic Location ID: e11034
Received 2020 Dec 14; Accepted 2021 Feb 9
Copyright: ©2021 Escribano et al.
Copyright year: 2021
Copyright holder: Escribano et al.
License: This is an open access article distributed under the terms of the Creative Commons Attribution License, which permits unrestricted use, distribution, reproduction and adaptation in any medium and for any purpose provided that it is properly attributed. For attribution, the original author(s), title, publication source (PeerJ) and either DOI or URL of the article must be cited.
License URL: https://creativecommons.org/licenses/by/4.0/

Keywords: Attitudes, Evidence-base practice, Health communication, Nursing students, Psychometrics

Funding: University of Alicante XARXES-I3CE-2018-4344 This study was funded by the Program Redes-I 3CE for Research in University Teaching of the Institute of Education Science (Vice-chancellorship of Quality and Educational Innovation) of the University of Alicante, edition 2018-19 (No. XARXES-I3CE-2018-4344). The funders had no role in study design, data collection and analysis, decision to publish, or preparation of the manuscript.

==============================
Background

Adequate communication skills in healthcare professionals are one of the key elements required for achieving high-quality healthcare. Thus, measurement instruments able to assess the dimensions related to these skills, including attitudes towards communication, are useful and convenient tools.

Objectives

To (a) cross-culturally adapt and validate a scale to measure attitudes towards communication in a sample of nursing students in the Spanish environment; (b) describe the perceived attitudes of nursing degree students towards communication.

Methods

We conducted an instrumental study. First, we adapted the scale by applying a standardised linguistic validation procedure. After that, we determined its structural equivalence and evaluated its psychometric properties.

Participants

A total of 255 students participated; their average age was 22.66 years (SD = 4.75) and 82% were female.

Results

The internal consistency of the scale was adequate (0.75), and the data fit well with the model (CFI = 0.99; TLI = 0.99; RMSEA = .01 95% CI [.00–.05]). The overall instrument score poorly correlated with the self-efficacy in communication skills variable.

Conclusions

The attitudes towards communication scores for these nursing students were high. The Spanish version of the Attitudes Towards Health Communication scale had adequate psychometric properties and this tool could quickly and easily be applied to assess the attitudes of health profession students.

Introduction

Changes in health systems have resulted in an increased focus on providing quality, patient-centred care, which must consider the transmission of information, participation in decision-making, and the respectful and responsible treatment of patients (Mata et al., 2019). This process prioritises the needs of patients and tries to improve the quality of their health and well-being as well as their satisfaction and the quality of their healthcare.

The literature identifies effective professional-patient communication as a key factor in obtaining positive results in terms of patient physical and emotional health (Shorey et al., 2018). Good communication improves patient-perceived satisfaction in healthcare processes and therefore also in the quality of their healthcare (Kourkouta & Papathanasiou, 2014); it also helps achieve greater adherence to prescribed treatments and compliance with recommended preventive activities (Holzinger et al., 2019). In contrast, inadequate communication is linked not only with lower satisfaction levels, but also with adverse events secondary to clinical errors (Guetterman et al., 2019).

Therefore, the acquisition of communication skills should be of special relevance in the training of health care professionals (Mata et al., 2019). However, the literature shows that extraordinarily little specific content on communication skills is currently included in their curricula (Ferrández-Antón et al., 2020; Ferreira-Padilla et al., 2015). Rather, the education of healthcare professionals tends to place a strong emphasis on the acquisition of technical skills to the detriment of nontechnical skills such as communication skills. These factors could explain the reports by nursing professionals that they do not feel sufficiently prepared in terms of communication skills (Foà et al., 2016), even though they perceive the learning of communication skills as an important and useful factor in professional practice (Coleman & McLaughlin, 2019).

Background

Communication skills can be acquired and improved through training (Cannity et al., 2020). Indeed, the European Higher Education Area (European Commission, 2005) is currently focusing its education on learning competencies which involve studying and integrating conceptual, attitudinal, and procedural dimensions into teaching (Tejada-Fernández & Ruiz-Bueno, 2016). Therefore, nursing curricula should not only include communication skills with an emphasis on theoretical content and skills acquisition but should also promote positive attitudes towards using these skills (Woloschuk, Harasym & Temple, 2004). As suggested in the literature, attitudes are a key component in the promotion of future behaviours (Sheeran et al., 2016). However, implementing a skills-based learning approach involves incorporating and developing new evaluation methodologies and resources and requires valid tools that are objective and reliable (MacLean et al., 2017) to assess attitudes towards communication skills in educational plans (Molinuevo & Torrubia, 2011).

Several instruments are available to assess attitudes towards communication in clinical practice. For example, the Nurses’ Attitudes Towards Communication with Patients Scale (ACO in its Spanish initialism; Espert & del Carmen, 2016) was developed to evaluate the attitudes of nursing students in terms of their behavioural, cognitive, and affective dimensions. However, the content of its items focused on hospitalisation situations applicable to certain specific nursing contexts; for example, the item “Information must be provided to the patient and/or their family about the nursing care being provided”. Another instrument, the Communication Skills Attitude Scale (CSAS; Rees, Sheard & Davies, 2002), is widely used internationally. The items included in this two-dimensional scale address situations commonly encountered by medical students, with items such as “In order to be a good doctor I must have good communication skills” or “Nobody is going to fail their medical degree for having poor communication skills”.

Finally, the Attitudes Towards Medical Communication Scale (Langille et al., 2001), is a short, one-dimensional scale which has been shown to have adequate psychometric properties. This instrument has several advantages compared to the other two: it is the shortest, a total construct score can be obtained from it, and the item content addresses general aspects of clinical communication skills that could be universally applicable to different health professionals. However, it has not yet been translated into Spanish or validated for use in this language.

Therefore, the objectives of this present study were to: (a) cross-culturally adapt and validate a scale to measure attitudes towards communication in a sample of nursing students in the Spanish environment; (b) describe the perceived attitudes of fourth-year nursing degree students towards communication.

Materials & Methods

Design

We conducted an instrumental study designed to develop an evaluation tool, which also included the generation or adaptation of new items and the examination of their psychometric properties (Carretero-Dios & Pérez, 2007). First, we adapted the scale into the Spanish context by applying a standardised linguistic validation procedure. After that, we determined the structural equivalence between the Spanish version and the original instrument and evaluated its psychometric properties.

Linguistic validation

We used a standard linguistic validation process (Acquadro et al., 2005) to obtain the Spanish version of the scale. First, two bilingual philologist translators whose mother tongue was Spanish, who were not linked to the project or study subject, and with experience in the field of health sciences translations/adaptations, translated the original English version into Spanish. Using both these direct translations, and after a consensus committee meeting, a first draft of the Spanish version was approved. The committee comprised two nurses with experience in adapting and validating scales in the healthcare context and a linguist who specialised in cross-cultural adaptation procedures.

Subsequently, two bilingual philologist translators whose mother tongue was English (one British and one American), who had no connection to the project or the subject of study and did not know the original English version of this questionnaire, independently translated the first Spanish consensus translation back into English in a blinded back-translation process. The committee reconvened to compare the two back-translations with the original English version and agreed upon a preliminary version in Spanish. This version was subsequently used to conduct the cognitive interviews.

The four translators were asked to assess the degree of difficulty of the direct and reverse translations of the items (on a scale of 1 to 10) and to classify the changes they had made into three categories (A, B, C). Type A changes were when no translation changes had been required and the sentence retained the same syntactic structure. Type B changes—morphosyntactic (B1), lexical-semantic (B2), or sociocultural (B3)—were changes that had been implemented to maintain equivalence. Items with type C changes were those where some of the items were not applicable in the context of the language into which it had been translated.

To examine the interpretability of the questionnaire, 8 cognitive interviews were carried out with nursing students; all of them were women and their mean age was 19.88 years (SD =1.25); 4 (50%) were first-year students and 4 were fourth-year students. All the participants completed the final translation back-translation consensus version of the questionnaire. The interviews were carried out by two trained nurses who were experts in conducting these types of interviews. Participants were asked to read the questionnaire aloud and to answer it while also indicating if they had understood the meaning of each item or if there were any unclear concepts. In addition, these participants were asked if they had understood the response alternatives and instructions, they had received prior to completing the questionnaire. Finally, they assessed the scale and they considered if any important items had been omitted.

Setting and sample

The data was collected between June and October 2019. The inclusion criteria were: (1) enrolment in a nursing course at the University of Alicante; (2) taking any fourth-year course; and (3) a fluent understanding of written and spoken Spanish. Students who had an Erasmus agreement who did not have an adequate knowledge of Spanish were excluded from the study. All the nursing students enrolled in a fourth-year nursing course subject at one of the abovenamed universities were invited to participate.

Of the total eligible participants (n = 400), a total of 255 students completed the questionnaire; thus, the response rate was 63.75%. Of the total sample, 52 students (20.39%) belonged to the 2018/19 academic year and a total of 203 students came from the 2019/20 academic year (79.61%).

Measures

An ad hoc questionnaire was developed to assess the sociodemographic variables of the participants, which included age, gender, nationality (Spanish or ‘other’), and marital status (single, married/cohabiting, separated/divorced, or widowed). In addition, these students were asked about any training in communication they had received during their nursing degree or in any other context.

To assess nursing students’ attitudes towards communication, linguistic validation of the Attitudes Towards Medical Communication Scale was used (Langille et al., 2001). This one-dimensional measurement instrument contains 12 items with a Likert-type response scale with 5 response options—from ‘strongly disagree’ (1) to ‘strongly agree’ (5). The total score ranges from 12 to 60 points, where the higher the score, the more positive the participants’ attitude towards communication. In its original version, this scale had an adequate internal consistency of 0.74.

To assess communication self-efficacy, we used the Spanish version (manuscript in progress) of the Self-Efficacy Questionnaire (SE-12; Axboe et al., 2016). This one-dimensional scale contains 12 items with a Likert-type response scale and 11 response options ranging from ‘very insecure’ (1) to ‘very confident’ (10); we also added the element ‘not relevant’. This scale was developed to assess the level of self-efficacy for the clinical communication skills acquired during nursing training courses. The total score ranges from 12 to 120 points, where the higher the score, the higher the students’ confidence in the communication skills they used with patients. The original version has a high internal consistency (′α = 0.95) and an adequate test-retest score (ICC =0.71).

Procedure

Details of this work were presented along with the request for informed consent to participation in this study. Participants were invited to participate through the internal communications system of the university institution (Virtual Campus). To maximise the response rate, we followed a standardised methodology and sent three reminders, sent one week apart from each other, which encouraged participation and provided the link to the questionnaire.

Data analysis

R software (version 3.4.0, R Core Team, 2017, 2014) was used in the confirmatory factor analysis (CFA) and internal consistency analysis. Scale performance, including floor and ceiling effects, was also analysed. The literature considers a floor or ceiling effect to have been reached when more than 15% of the participants in the lower and upper range, respectively, responded (Lim et al., 2015). Following the criteria published by Rhemtulla, Brosseau-Liard & Savalei (2012), we considered our data to be ordinal. The robust weighted least squares means and variance (WLSMV) estimation method from the Lavaan package was used for the analysis of the ordinal data (Rhemtulla, Brosseau-Liard & Savalei, 2012).

Three indices were considered in the analysis of the data fit to the model: the comparative fit index (CFI), Tucker–Lewis index (TLI), and root mean square error of approximation (RMSEA). Values exceeding 0.90 were considered adequate for the CFI and TLI indicators (Bentler, 1990; Hu & Bentler, 1999). For the RMSEA index, values less than 0.5 indicated an adequate fit (Brownie, 1993; (Green & Yang, 2009)).

The ordinal alpha coefficient, considering the estimated model, was used to analyse the internal consistency of the instrument, and was considered more accurate for the categorical response scales (Zumbo, Gadermann & Zeisser, 2007). This measure is considered an unbiased estimator of internal consistency as long as the essentially tau-equivalent model fits the data (McDonald, 1999). The self-efficacy in communication skills variable was used to assess the construct validity of the instrument using the Pearson correlation coefficient (r). We required our results to meet the consensus-based standards for the selection of health measurement instruments (COSMIN; Mokkink et al., 2017) recommended correlations (between 0.3 and 0.5) because these instruments measured different but related constructs. Finally, a descriptive analysis (means and SDs) was performed for the scale for the perceived attitude towards communication in nursing students.

Ethical issues

This study followed the criteria established by the Declaration of Helsinki and the European Union Standards of Good Clinical Practice and obtained approval by the Bioethics Committee at the University of Alicante (reference number: UA-2018-10-24). All the participants received information about the voluntary nature of their participation, the possibility of withdrawal of their consent at any time, and the planned treatment of the collected data. All the data obtained was treated confidentially and used exclusively for research purposes.

Results

Linguistic validation

Translation-back-translation process

The average degree of difficulty of the translation ranged from 2 (item 5) to 6 (item 1) on a scale of 1 to 8. The average difficulty of the back-translation ranged from 1.5 (items 1 and item 5) to 8.5 (item 8) on a scale of 1 to 9. In both the translation and back-translation, most of the changes required were of the morphosyntactic (B1) or lexical-semantic (B2) type. No item required type C changes. Of note, the translators did not agree on the difficulty level of items 3 and 6 in the translation or in the back-translation.

Cognitive interviews

Participants experienced comprehension difficulties for items 3, 6, 9, and 10. Item 3 was exceedingly long and difficult to understand and so, it required morphosyntactic (B1) and lexical-semantic (B2) modifications. The difficulty for the remaining items (which also required B1 and B2 changes) was associated with the use of specific expressions. For item 6 we solved this by changing “...the patient experience...” to “...the patients’ experience of their disease”. In the case of item 9, we clarified the term “...psychosocial problems...” by adding some examples, “...psychosocial problems (for example, family or economic problems, etc.)”. For item 10, the expression “...health results...” was made clearer by changing it to “...health results (for example, readmissions, quality of life, etc.)”.

Evaluation of psychometric properties

Sample sociodemographic data

Table 1 shows the sociodemographic data for the 255 participants, of which, 82% were women (n = 209). The mean age of the cohort was 22.66 years (SD =4.75), with an age range between 18 and 48 years.

Table 1 Socio-demographic characteristics of participants (N = 255).

Variable	N (%)	
Female	209 (82)	
Age (M/SD)	22.66 (2.41)	
Nationality		
Spanish	248 (97.30)	
Other	7 (2.7)	
Marital status		
Single	241 (94.5)	
Married/Cohabited	12 (4.7)	
Separated/Divorced	2 (0.80)	
Cohort		
2018/2019	52 (20.39)	
2019/2020	203 (79.61)	
Previous Grade communication skills training	
Yes	239 (93.70)	
No	16 (6.30)	
Previous communication skills training in other context	
Yes	42 (16.50)	
No	213 (83.50)	
Self.Efficacy towards communication (M/SD)a	83.09 (13.70)	
Notes.

M Mean

SD Standard deviation

a This analysis was performed with 203 participants.

Scale performance and psychometric properties

The scale performance results are shown in Table 2; ceiling effects were observed for all the items included in the scale. The CFA showed that the data fit adequately to the original structure (CFI =0.96; TLI =0.95; RMSEA =0.03 [95% CI [0.00–0.05]]) (Table 3). The estimated factor loads for most of the items varied from 0.24 to 0.72, except for item 3, which showed a factor load of 0.04. Elimination of item 3, “In healthcare provision teams, cohesion is desirable, but one professional alone can do little to promote it”, improved the data fit (CFI =0.99; TLI =0.99; RMSEA =0.01 95% CI [0.00–0.05]). The estimated factor loads of the items varied from 0.24 to 0.72 (Fig. 1). Lastly, because the measures were essentially tau-equivalent (i.e., all the items had the same factor load), we tested a more restricted estimation model and found that the data continued to fit the model, although with a slightly poorer parameter results (CFI =0.93; TLI =0.93; RMSEA =0.04 [95% CI [0.004–.06]]).

Table 2 Performance of the scale.

Item	Min	Max	M (SD)	Skewness	Kurtosis	Floor effectn (%)	Ceiling effectn (%)	
Item 1	3	5	4.87 (0.37)	−2.83	7.78	0 (0)	224 (87.80)	
Item 2	1	5	4.75 (0.84)	−3.80	13.87	10 (3.90)	225 (88.20)	
Item 3	1	5	3.28 (1.26)	−.30	−.95	28 (11)	48 (18.80)	
Item 4	1	5	4.75 (0.51)	−3.55	16.91	1 (0.4)	215 (84.30)	
Item 5	3	5	4.95 (0.25)	−4.71	23.75	0 (0)	242 (94.90)	
Item 6	1	5	4.75 (0.60)	−3.59	17.11	3 (1.2)	205 (80.4)	
Item 7	1	5	4.65 (0.57)	−1.93	6.01	1 (0.4)	176 (69)	
Item 8	3	5	4.76 (0.46)	−1.58	1.39	0 (0)	196 (76.9)	
Item 9	3	5	4.69 (0.58)	−1.69	1.81	0 (0)	190 (74.5)	
Item 10	3	5	4.83 (0.40)	−2.11	3.53	0 (0)	213 (83.5)	
Item 11	1	5	4.77 (0.64)	−3.98	18.84	4 (1.6)	212 (83.1)	
Item 12	1	5	4.80 (0.50)	−3.51	17.08	1 (0.4)	213 (83.5)	
Notes.

M Mean

SD Standard deviation

Min Minimum

Max Maximum

Table 3 Confirmatory analysis and internal reliability consistency (N = 255).

Fitted model	Chi-square	p	gl	CFI	TLI	RMSEA [95%CI]	Ordinal alphac	
Originala	66.151	0.12	54	0.96	0.95	0.03 (.00–.05)	0.72	
Modifiedb	45.900	0.39	44	0.99	0.99	0.01 (.00–.05)	0.75	
Notes.

a Original proposed by Langille et al. (2001).

b Modified, without ”Item 3”; CFI, Comparative fix index; TLI, Tucker-Lewis index; RMSEA, Root mean square error of approximation; CI, Confident interval.

c Ordinal Alpha calculated across Structural Equation Model (SEM).

Figure 1 Confirmatory factor analysis.

Confirmatory factor analysis without item 3 graph extracted via the Lavaan package in the R freeware; AHC = Attitudes towards Health Communication, Spanish version of Attitudes towards Medical Communication Scale (Langille et al., 2001).

The internal consistency, calculated with the alpha ordinal, and considering the specified model after eliminating item 3, was 0.75. Furthermore, to assess the construct validity, the total scores obtained from the attitude and self-efficacy towards communication poorly correlated but were statistically significant (r =0.21, p = .003; n = 203).

Descriptive data for the participants’ attitudes towards communication skills

The overall scale score ranged between 43 and 55, with a mean of 52.62 (SD =2.42). Statistically significant differences according to gender were found in the overall scores, where women obtained a higher mean score (M =52.89; SD =2.13) compared to men (M = 51.35; SD =3.16; Mann–Whitney U = 3.453, p =0.002). No significant differences were found between the 2018/2019 and 2019/2010 cohorts (Mann–Whitney U = 4.861; p =0.37).

Discussion

The purpose of this present study was to validate the Spanish translation of the Attitudes Towards Medical Communication Scale (Langille et al., 2001) in a sample of nursing students in the Spanish setting and to describe the attitudes perceived by nursing-degree students. Our findings indicated that the Spanish version of the scale, Attitude Towards Health Communication (AHC-S), called the ‘Actitudes de la Comunicación Sanitaria’ scale in Spanish (see Supplemental Files) is a valid and reliable instrument to assess the attitudes of nursing-degree students towards communication. In addition, this scale has also been validated in a health context different from the one in which it was originally developed (for use in medical students). Therefore, we concluded that this is a valid scale for the extensive analysis of attitudes towards communication in other professional healthcare contexts.

The CFA showed that the scale maintained its original structure and had adequate adjustment indices when used to assess Spanish nursing students. However, given the difficulty in understanding item 3 in the cognitive interviews we performed, as well as the low factor loading of this item in the CFA, we omitted item 3 and modified the final Spanish version of the scale to contain only 11 items. Future research will be needed to check the adjustment of the data in other contexts to confirm the original structure and the operation of item 3.Following recommendations based on current evidence (Black et al., 2014; Yang & Green, 2011), the internal consistency was initially adjusted based on the specifications in the structural equation modelling. The internal consistency was adequate (>0.70; George & Mallery, 2003) and was similar to that obtained in the original version of the scale (Langille et al., 2001).

In terms of construct validity, we found a weak positive correlation (Cohen, 1992) between the attitudes towards communication score and the SE-12 scale (Axboe et al., 2016), although this factor was not evaluated in the original version (Langille et al., 2001). The literature indicates that there is a relationship between these two variables in the educational field and that they are both relevant in professional practice (Sung, Huang & Lin, 2015) and future behaviour (Sheeran et al., 2016). However, a recent study by our team (Juliá-Sanchis et al., 2020) observed only moderate correlations (r =0.35) between perceived abilities in communication skills and scores on the Attitudes Towards Communication Scale, including on the 12 items in the original version. Thus, we recommend continuing the analysis of the construct validity of this scale in future lines of research.

Our results showed that the items on this scale had a ceiling effect (Lim et al., 2015), which suggests that it might not evaluate a wide range of the potential variability among students who have positive attitudes towards communication, at least for samples with similar characteristics to the cohort considered in this study. However, it should be noted that the previous communication training of these students was mostly received as part of their formal compulsory training. Thus, the fact that previous studies highlight specific training as an influencing factor in attitudinal improvement (Lichtenstein et al., 2018) could explain their high scores in attitudes towards communication.

In terms of a general description of the attitudes of nursing students towards communication, our results showed that high scores were as high as those shown in other national studies in students of other health professions (Molinuevo et al., 2016), and specifically among those in nursing contexts (Molinuevo & Torrubia, 2011). Similar results were also found in international studies in medical students (Langille et al., 2001) and among nursing professionals (Cronin & Finn, 2017; Panczyk et al., 2019). Therefore, it seems that students have positive attitudes towards communication, regardless of the type of healthcare profession they work in and the environment in which they live. Nonetheless, much less work has been done in nursing students compared to similar studies in medical or dental students (Abdrbo, 2017). Hence, this current study helps to increase and update our knowledge about this issue in nursing students. Furthermore, these results have important implications within the European Higher Education Area given the emphasis this body places on the acquisition and evaluation of attitudinal competences in the learning-teaching process (European Commission, 2005).

Regarding the sociodemographic factors of this cohort, as reported elsewhere in the literature (Molinuevo & Torrubia, 2011; Koponen, Pyörälä & Isotalus, 2012), we also detected gender-related differences in our cohort. Previous studies have revealed that male students had greater difficulty in learning communication skills and that this may be because the acquisition of these skills requires emotional exposure, which can be more characteristic of women (Loffler-Stastka et al., 2016). Thus, according to this previous work, nursing schools face a major challenge when trying to improve attitudes towards better acquisition of communication skills among male students (Gude et al., 2020). However, more studies with a gender perspective (Lichtenstein et al., 2018) will be needed to fully assess whether there are indeed currently gender-based differences in nursing, and to analyse what factors influence attitudes towards communication so that these can be considered when providing student training.

Limitations

Firstly, we used the natural opportunity presented by the enrolment of nursing students into their courses to obtain a convenience sample. Although participation was always voluntary and did not influence their evaluation outcomes, this may have compromised the external validity of our results. Therefore, the structure of this scale should be confirmed with representative and randomised nursing student samples at the national level to guarantee the that these results can be generalised. Furthermore, the validity of the Spanish version of this scale only applies to undergraduate nursing students. Therefore, if it is to be used in other sample cohorts such as postgraduate students or clinical healthcare professionals, including nursing professionals, evidence for the validity of this scale must first be obtained for these groups.

Finally, the overall participation rate in this work was moderate, and specifically, was low in the 2018/2019 cohort and so future research should aim to increase these participation levels. One reason for this may have been the timing of our data collection, which was preferable in the 2019/2020 cohort because these students were starting the academic year. Therefore, the timing of the data collection for the 2019/2020 period corresponded to the time in which the theoretical part of the course was taught, and when the students were more strongly linked to the university. In contrast, in the 2018/2019 cohort, the students were surveyed at the end of the academic year when they were already engaged in their clinical practices in hospitals and had a weaker association with the university. Nevertheless, this characteristic had no impact on the internal validity of the study because no differences were found in the attitude towards communication scores variable that we were specifically focusing on in this current validation study.

Conclusions

The AHC-S is valid and reliable tool for assessing the attitudes of nursing degree students; it represents a useful tool for university professors in the assessment of health professions given that it is self-administered, quick and easy to complete, and helps provide useful data for deciding whether improvements or changes in training programs are required when low levels of certain attitudes are identified. Future lines of research should be aware of the ceiling effect of the scale and should aim to validate this tool with students and health professionals in other contexts.

Supplemental Information

Supplemental Information 1 Blinded dataset

Click here for additional data file.

Supplemental Information 2 Spanish version scale

Click here for additional data file.

Additional Information and Declarations

Competing Interests

Author Contributions

Human Ethics

Data Availability

The authors declare there are no competing interests.

Silvia Escribano, Rocío Juliá-Sanchis and María José Cabañero-Martínez conceived and designed the experiments, performed the experiments, analyzed the data, prepared figures and/or tables, authored or reviewed drafts of the paper, and approved the final draft.

Sofía García-Sanjuán analyzed the data, authored or reviewed drafts of the paper, and approved the final draft.

Nereida Congost-Maestre performed the experiments, authored or reviewed drafts of the paper, and approved the final draft.

The following information was supplied relating to ethical approvals (i.e., approving body and any reference numbers):

This study was approved by the Bioethics Committee at the University of Alicante (reference number: UA-2018-10-24).

The following information was supplied regarding data availability:

Raw data are available in the Supplementary Files.

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
