# Peer review of "Psychometric properties of the Attitudes towards Medical Communication Scale in nursing students"

_PeerJ, doi:10.7717/peerj.11034_

## Round 0.1 · original submission · Minor Revisions

The manuscript has been carefully reviewed by two external reviewers and myself. Before proceeding to recommend its publication, it is necessary that the authors address the minor comments raised by the reviewers.

Reviewer 1 ·

Basic reporting

The authors have been very conscientious and systematic in their preparation, writing and reporting the results of the cross-cultural validation and adaptation of a scale.
The introduction is well founded. However, in the bacground section, you should have contemplated and cited other validated scales in Spanish in your immediate area. For example The Communication Skills Scale (CSS) (Leal-Costa et al., 2019; Leal-Costa et al., 2016).

Experimental design

The method is well explained and contains all the sections that a validation research and adaptation of a questionnaire must have.
The results have been flawlessly displayed. Finally, there are 11 items in the Spanish version and I sincerely believe that your great work is credited.

Validity of the findings

I only must make a final reflection on the meaning of your work. The Attitudes Towards Medical Communication Scale was initially validated in medical students. Later it has been used in health professionals in various fields and in students as well.
I think, to be honest, the external validity of your validation should be improved. For this, it would be necessary to use a representative sample of the Spanish population to which the scale is targeted. Not only fourth year nursing students in the southeast of Spain. I mean that validation should be considered in a multiprofessional sample (mainly postgraduate doctors and nurses). I believe that students have limited access to communication with patients. Communication is reduced to their limited clinical practices and their clinical skills are not sufficiently developed, leaving non-clinical skills in the background. But this is just a reflection that should be considered when naming the scale in Spanish (indicating that its validity includes only nursing students or that the population it is addressed to are students). However, I think this would be a target for further research, so I only suggest that you include these arguments in the limitations section.

Additional comments

Thank you very much for giving me the opportunity to review this excellent work. It is a very professionally written manuscript with an impeccable scientific structure.
The authors have been very conscientious and systematic in their preparation, writing and reporting the results of the cross-cultural validation and adaptation of a scale.
The introduction is well founded. However, in the bacground section, you should have contemplated and cited other validated scales in Spanish in your immediate area. For example The Communication Skills Scale (CSS) (Leal-Costa et al., 2019; Leal-Costa et al., 2016).
The method is well explained and contains all the sections that a validation research and adaptation of a questionnaire must have.
The results have been flawlessly displayed. Finally, there are 11 items in the Spanish version and I sincerely believe that your great work is credited.
I only must make a final reflection on the meaning of your work. The Attitudes Towards Medical Communication Scale was initially validated in medical students. Later it has been used in health professionals in various fields and in students as well.
I think, to be honest, the external validity of your validation should be improved. For this, it would be necessary to use a representative sample of the Spanish population to which the scale is targeted. Not only fourth year nursing students in the southeast of Spain. I mean that validation should be considered in a multiprofessional sample (mainly postgraduate doctors and nurses). I believe that students have limited access to communication with patients. Communication is reduced to their limited clinical practices and their clinical skills are not sufficiently developed, leaving non-clinical skills in the background. But this is just a reflection that should be considered when naming the scale in Spanish (indicating that its validity includes only nursing students or that the population it is addressed to are students).
However, I think this would be a target for further research, so I only suggest that you include these arguments in the limitations section.

·

Basic reporting

See general comments

Experimental design

See general comments

Validity of the findings

See general comments

Additional comments

The manuscript by Escribano et al describes the authors’ attempt to translate the Attitude Toward Communication Scale for use among nursing students in Spain. The authors seem to have a good level of sophistication about the requirements of creating a translation and demonstrating its psychometric characteristics, and the paper therefore contributes to the literature in disseminating a tool for use in a culture and in a health profession for which it was not previously available. Although generally favorable toward this paper, I came across several instances of sentences that did not make sense, and it appeared that they more related to proofreading problems than English-Spanish translation. This should not have happened.
There were several issues, some larger, some smaller, where I thought the paper could be improved:
• The authors refer to the fact that there seems to be little skills training in communication, referring to nursing, for which I agree. However, their references largely relate to medicine. They should be more careful about being sure to cite nursing literature if they want to make a case about nursing.
• As noted above, there are some issues about meaning. Line 75: In “This evidence-based practice approach…” it’s not clear what this refers to. Lines 129-130, something is missing. Line 152—what additional study questions? Line 228: the sentence is missing something.
• The authors used two study sites, one with a very high response rate the other very low. Nowhere did I see any explanation for this or a concern for combining the two sites’ data together. In fact, given the very small n in the second site, I would recommend that the authors demonstrate that the data were comparable across sites in order to combine the two, or even better, that they simply delete the small number from the second site in their analyses.
• Why do the authors never show us he items? Referring to them by number when the reader is totally in the dark as to their actual content is a disservice. Especially when we come to the cognitive interviews, we are lost without knowing what the items were.
• Does the fact that there were ceiling effects for many of the items not concern the authors? This needs further discussion.
• In terms of validity evidence, the authors rush through the fact that their scale was moderately correlated with the Self-Efficacy scale. This seems like very limited validity evidence, very weakly explained. If the authors want to spend so much time providing information on their fancier statistics, why do they not spend time talking about types of validity and what their moderate correlation with another validated scale means.
• On lines 296-297, the authors make reference to the Theory of Planned Behavior in a way that is both loose and inaccurate. If they want to give their paper some conceptual/theoretical backing, they should do so with far more care. Otherwise, it is simply window dressing for those who will be impressed by the simple mention of the theory.
• The authors report gender differences. This is good and these findings are consistent with lots of other findings in the literature. It’s surprising that they don’t tell us about the scale scores in relationship to training in communications within their sample, nor do they tell is anything about their scores in relationship to those found in other cultures and in other health professions. This would make the paper far richer.
• The authors mention notable limitations in a couple of sentences and then simply move on, as if mentioning these was sufficient. Once we are aware of the limitations, the authors are obligated to inform readers what to make of these limitations and/or how they might be addressed or overcome in the future.
In general, this is a solid paper, done by people who clearly know their way around translation and back-translation issues and who hove good statistical expertise among their team and consultants. It is my feeling that the improvements suggested here could be made without a major re-write, and that these improvements would increase its contribution significantly.

---

## Round 0.2 · accepted · Accept

I thank the authors for their efforts in reviewing the first version of the manuscript.